# Perspective Transformer Nets: Learning Single-View 3D Object Reconstruction without 3D Supervision

**Xinchen Yan**[1]    **Jimei Yang**[2]    **Ersin Yumer**[2]    **Yijie Guo**[1]    **Honglak Lee**[1,3]

[1]University of Michigan, Ann Arbor
[2]Adobe Research
[3]Google Brain

{xcyan,guoyijie,honglak}@umich.edu, {jimyang,yumer}@adobe.com

## Abstract

Understanding the 3D world is a fundamental problem in computer vision. However, learning a good representation of 3D objects is still an open problem due to the high dimensionality of the data and many factors of variation involved. In this work, we investigate the task of single-view 3D object reconstruction from a learning agent's perspective. We formulate the learning process as an interaction between 3D and 2D representations and propose an encoder-decoder network with a novel projection loss defined by the perspective transformation. More importantly, the projection loss enables the unsupervised learning using 2D observation without explicit 3D supervision. We demonstrate the ability of the model in generating 3D volume from a single 2D image with three sets of experiments: (1) learning from single-class objects; (2) learning from multi-class objects and (3) testing on novel object classes. Results show superior performance and better generalization ability for 3D object reconstruction when the projection loss is involved.

## 1 Introduction

Understanding the 3D world is at the heart of successful computer vision applications in robotics, rendering and modeling [19]. It is especially important to solve this problem using the most convenient visual sensory data: 2D images. In this paper, we propose an end-to-end solution to the challenging problem of predicting the underlying true shape of an object given an arbitrary single image observation of it. This problem definition embodies a fundamental challenge: Imagery observations of 3D shapes are interleaved representations of intrinsic properties of the shape itself (e.g., geometry, material), as well as its extrinsic properties that depend on its interaction with the observer and the environment (e.g., orientation, position, and illumination). Physically principled shape understanding should be able to efficiently disentangle such interleaved factors.

This observation leads to insight that an end-to-end solution to this problem from the perspective of learning agents (neural networks) should involve the following properties: 1) the agent should understand the physical meaning of how a 2D observation is generated from the 3D shape, and 2) the agent should be conscious about the outcome of its interaction with the object; more specifically, by moving around the object, the agent should be able to correspond the observations to the viewpoint change. If such properties are embodied in a learning agent, it will be able to disentangle the shape from the extrinsic factors because these factors are trivial to understand in the 3D world. To enable the agent with these capabilities, we introduce a built-in camera system that can transform the 3D object into 2D images in-network. Additionally, we architect the network such that the latent representation disentangles the shape from view changes. More specifically, our network takes as input an object image and predicts its volumetric 3D shape so that the perspective transformations of predicted shape match well with corresponding 2D observations.

We implement this neural network based on a combination of image encoder, volume decoder and perspective transformer (similar to spatial transformer as introduced by Jaderberg et al. [6]). During training, the volumetric 3D shape is gradually learned from single-view input and the feedback of other views through back-propagation. Thus at test time, the 3D shape can be directly

generated from a single image. We conduct experimental evaluations using a subset of 3D models from ShapeNetCore [1]. Results from single-class and multi-class training demonstrate excellent performance of our network for volumetric 3D reconstruction. Our main contributions are summarized below.

- We show that neural networks are able to predict 3D shape from single-view without using the ground truth 3D volumetric data for training. This is made possible by introducing a 2D silhouette loss function based on perspective transformations.
- We train a single network for multi-class 3D object volumetric reconstruction and show its generalization potential to unseen categories.
- Compared to training with full azimuth angles, we demonstrate comparatively similar results when training with partial views.

## 2    Related Work

**Representation learning for 3D objects.**    Recently, advances have been made in learning deep neural networks for 3D objects using large-scale CAD databases [22, 1]. Wu et al. [22] proposed a deep generative model that extends the convolutional deep belief network [11] to model volumetric 3D shapes. Different from [22] that uses volumetric 3D representation, Su et al. [18] proposed a multi-view convolutional network for 3D shape categorization with a view-pooling mechanism. These methods focus more on 3D shape recognition instead of 3D shape reconstruction. Recent work [20, 14, 4, 2] attempt to learn a joint representation for both 2D images and 3D shapes. Tatarchenko et al. [20] developed a convolutional network to synthesize unseen 3D views from a single image and demonstrated the synthesized images can be used them to reconstruct 3D shape. Qi et al. [14] introduced a joint embedding by combining volumetric representation and multi-view representation together to improve 3D shape recognition performance. Girdhar et al. [4] proposed a generative model for 3D volumetric data and combined it with a 2D image embedding network for single-view 3D shape generation. Choy et al. [2] introduce a 3D recurrent neural network (3D-R2N2) based on long-short term memory (LSTM) to predict the 3D shape of an object from a single view or multiple views. Compared to these single-view methods, our 3D reconstruction network is learned end-to-end and the network can be even trained without ground truth volumes.

Concurrent to our work, Renzede et al. [16] introduced a general framework to learn 3D structures from 2D observations with 3D-2D projection mechanism. Their 3D-2D projection mechanism either has learnable parameters or adopts non-differentiable component using MCMC, while our perspective projection nets is both differentiable and parameter-free.

**Representation learning by transformations.**    Learning from transformed sensory data has gained attention  [12, 5, 15, 13, 23, 6, 24] in recent years. Memisevic and Hinton [12] introduced a gated Boltzmann machine that models the transformations between image pairs using multiplicative interaction. Reed et al. [15] showed that a disentangled hidden unit representations of Boltzmann Machines (disBM) could be learned based on the transformations on data manifold. Yang et al. [23] learned out-of-plane rotation of rendered images to obtain disentangled identity and viewpoint units by curriculum learning. Kulkarni et al. [9] proposed to learn a semantically interpretable latent representation from 3D rendered images using variational auto-encoders [8] by including specific transformations in mini-batches. Complimentary to convolutional networks, Jaderberg et al. [6] introduced a differentiable sampling layer that directly incorporates geometric transformations into representation learning. Concurrent to our work, Wu et al. [21] proposed a 3D-2D projection layer that enables the learning of 3D object structures using 2D keypoints as annotation.

## 3    Problem Formulation

In this section, we develop neural networks for reconstructing 3D objects. From the perspective of a learning agent (e.g., neural network), a natural way to understand one 3D object $X$ is from its 2D views by transformations. By moving around the 3D object, the agent should be able to recognize its unique features and eventually build a 3D mental model of it as illustrated in Figure 1(a). Assume that $I^{(k)}$ is the 2D image from $k$-th viewpoint $\alpha^{(k)}$ by projection $I^{(k)} = P(X; \alpha^{(k)})$, or rendering in graphics. An object $X$ in a certain scene is the entanglement of shape, color and texture (its intrinsic properties) and the image $I^{(k)}$ is the further entanglement with viewpoint and illumination (extrinsic parameters). The general goal of understanding 3D objects can be viewed as disentangling intrinsic properties and extrinsic parameters from a single image.

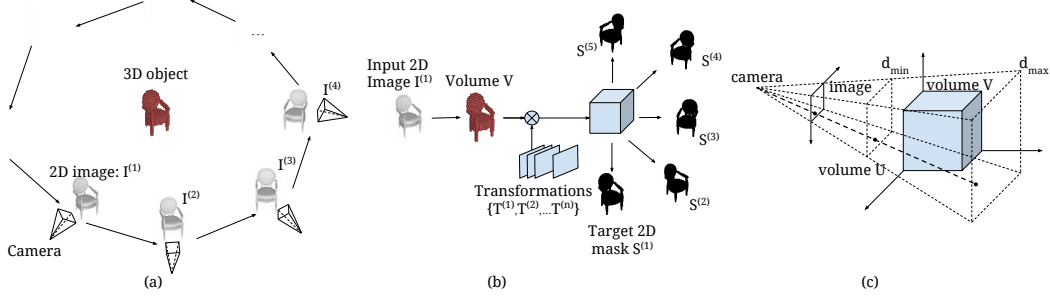

Figure 1: (a) Understanding 3D object from learning agent's perspective; (b) Single-view 3D volume reconstruction with perspective transformation. (c) Illustration of perspective projection. The minimum and maximum disparity in the screen coordinates are denoted as $d_{min}$ and $d_{max}$.

In this paper, we focus on the 3D shape learning by ignoring the color and texture factors, and we further simplify the problem by making the following assumptions: 1) the scene is clean white background; 2) the illumination is constant natural lighting. We use the volumetric representation of 3d shape $\mathbf{V}$ where each voxel $\mathbf{V}_i$ is a binary unit. In other words, the voxel equals to one, i.e., $\mathbf{V}_i = 1$, if the $i$-th voxel sapce is occupied by the shape; otherwise $\mathbf{V}_i = 0$. Assuming the 2D silhouette $S^{(k)}$ is obtained from the $k$-th image $I^{(k)}$, we can specify the 3D-2D projection $S^{(k)} = P(\mathbf{V}; \alpha^{(k)})$. Note that 2D silhouette estimation is typically solved by object segmentation in real-world but it becomes trivial in our case due to the white background.

In the following sub-sections, we propose a formulation for learning to predict the volumetric 3D shape $\mathbf{V}$ from an image $I^{(k)}$ with and without the 3D volume supervision.

## 3.1 Learning to Reconstruct Volumetric 3D Shape from Single-View

We consider single-view volumetric 3D reconstruction as a dense prediction problem and develop a convolutional encoder-decoder network for this learning task denoted by $\hat{\mathbf{V}} = f(I^{(k)})$. The encoder network $h(\cdot)$ learns a *viewpoint-invariant* latent representation $h(I^{(k)})$ which is then used by the decoder $g(\cdot)$ to generate the volume $\hat{\mathbf{V}} = g(h(I^{(k)}))$. In case the ground truth volumetric shapes $\mathbf{V}$ are available, the problem can be easily considered as learning volumetric 3D shapes with a regular reconstruction objective in 3D space: $\mathcal{L}_{vol}(I^{(k)}) = ||f(I^{(k)}) - \mathbf{V}||_2^2$.

In practice, however, *the ground truth volumetric 3D shapes may not be available for training*. For example, the agent observes the 2D silhouette via its built-in camera without accessing the volumetric 3D shape. Inspired by the space carving theory [10], we propose a silhouette-based volumetric loss function. In particular, we build on the premise that a 2D silhouette $\hat{S}^{(j)}$ projected from the generated volume $\hat{\mathbf{V}}$ under certain camera viewpoint $\alpha^{(j)}$ should match the ground truth 2D silhouette $S^{(j)}$ from image observations. In other words, if all the generated silhouettes $\hat{S}^{(j)}$ match well with their corresponding ground truth silhouettes $S^{(j)}$ for all $j$'s, then we hypothesize that the generated volume $\hat{\mathbf{V}}$ should be as good as one instance of *visual hull* equivalent class of the ground truth volume $\mathbf{V}$ [10]. Therefore, we formulate the learning objective for the $k$-th image as

$$\mathcal{L}_{proj}(I^{(k)}) = \sum_{j=1}^{n} \mathcal{L}_{proj}^{(j)}(I^{(k)}; S^{(j)}, \alpha^{(j)}) = \frac{1}{n} \sum_{j=1}^{n} ||P(f(I^{(k)}); \alpha^{(j)}) - S^{(j)}||_2^2, \tag{1}$$

where $j$ is the index of output 2D silhouettes, $n$ is the number of silhouettes used for each input image and $P(\cdot)$ is the 3D-2D projection function. Note that the above training objective Eq. (1) enables training without using ground-truth volumes. The network diagram is illustrated in Figure 1(b). A more general learning objective is given by a combination of both objectives:

$$\mathcal{L}_{comb}(I^{(k)}) = \lambda_{proj}\mathcal{L}_{proj}(I^{(k)}) + \lambda_{vol}\mathcal{L}_{vol}(I^{(k)}), \tag{2}$$

where $\lambda_{proj}$ and $\lambda_{vol}$ are constants that control the tradeoff between the two losses.

## 3.2 Perspective Transformer Nets

As defined previously, 2D silhouette $S^{(k)}$ is obtained via perspective projection given input 3D volume $\mathbf{V}$ and specific camera viewpoint $\alpha^{(k)}$. In this work, we implement the perspective projection

(see Figure 1(c)) with a 4-by-4 transformation matrix $\mathbf{\Theta}_{4\times4}$, where $\mathbf{K}$ is camera calibration matrix and $(\mathbf{R}, \mathbf{t})$ is extrinsic parameters.

$$\mathbf{\Theta}_{4\times4} = \begin{bmatrix} \mathbf{K} & 0 \\ 0^T & 1 \end{bmatrix} \begin{bmatrix} \mathbf{R} & \mathbf{t} \\ 0^T & 1 \end{bmatrix} \tag{3}$$

For each point $\mathbf{p}_i^s = (x_i^s, y_i^s, z_i^s, 1)$ in 3D world coordinates, we compute the corresponding point $\mathbf{p}_i^t = (x_i^t, y_i^t, 1, d_i^t)$ in screen coordinates (plus disparity $d_i^t$) using the perspective transformation: $\mathbf{p}_i^s \sim \mathbf{\Theta}_{4\times4}\mathbf{p}_i^t$.

Similar to the spatial transformer network introduced in [6], we propose a 2-step procedure : (1) performing dense sampling from input volume (in 3D world coordinates) to output volume (in screen coordinates), and (2) flattening the 3D spatial output across disparity dimension. In the experiment, we assume that transformation matrix is always given as input, parametrized by the viewpoint $\alpha$. Again, the 3D point $(x_i^s, y_i^s, z_i^s)$ in input volume $\mathbf{V} \in \mathbb{R}^{H \times W \times D}$ and corresponding point $(x_i^t, y_i^t, d_i^t)$ in output volume $\mathbf{U} \in \mathbb{R}^{H' \times W' \times D'}$ is linked by perspective transformation matrix $\mathbf{\Theta}_{4\times4}$. Here, $(W, H, D)$ and $(W', H', D')$ are the width, height and depth of input and output volume, respectively.

We summarize the dense sampling step and channel-wise flattening step as follows.

$$U_i = \sum_n^H \sum_m^W \sum_l^D V_{nml} \max(0, 1 - |x_i^s - m|) \max(0, 1 - |y_i^s - n|) \max(0, 1 - |z_i^s - l|) \tag{4}$$

$$S_{n'm'} = \max_{l'} U_{n'm'l'}$$

Here, $U_i$ is the $i$-th voxel value corresponding to the point $(x_i^t, y_i^t, d_i^t)$ (where $i \in \{1, ..., W' \times H' \times D'\}$). Note that we use the $\max$ operator for projection instead of summation along one dimension since the volume is represented as a binary cube where the solid voxels have value 1 and empty voxels have value 0. Intuitively, we have the following two observations: (1) each empty voxel will not contribute to the foreground pixel of $S$ from any viewpoint; (2) each solid voxel can contribute to the foreground pixel of $S$ only if it is visible from specific viewpoint.

### 3.3 Training

As the same volumetric 3D shape is expected to be generated from different images of the object, the encoder network is required to learn a 3D view-invariant latent representation

$$h(I^{(1)}) = h(I^{(2)}) = \cdots = h(I^{(k)}) \tag{5}$$

This sub-problem itself is a challenging task in computer vision [23, 9]. Thus, we adopt a two-stage training procedure: first we learn the encoder network for a 3D view-invariant latent representation $h(I)$ and then train the volumetric decoder with perspective transformer networks. As shown in [23], a disentangled representation of 2D synthetic images can be learned from consecutive rotations with a recurrent network, we pre-train the encoder of our network using a similar curriculum strategy so that the latent representation only contains 3D view-invariant identity information of the object. Once we obtain an encoder network that recognizes the identity of single-view images, we next learn the volume generator regularized by the perspective transformer networks. To encourage the volume decoder to learn a consistent 3D volume from different viewpoints, we include the projections from neighboring viewpoints in each mini-batch so that the network has relatively sufficient information to reconstruct the 3D shape.

## 4 Experiments

**ShapeNetCore.** This dataset contains about 51,300 unique 3D models from 55 common object categories [1]. Each 3D model is rendered from 24 azimuth angles (with steps of $15°$) with fixed elevation angles ($30°$) under the same camera and lighting setup. We then crop and rescale the centering region of each image to $64 \times 64 \times 3$ pixels. For each ground truth 3D shape, we create a volume of $32 \times 32 \times 32$ voxels from its canonical orientation ($0°$).

**Network Architecture.** As shown in Figure 2, our encoder-decoder network has three components: a 2D convolutional encoder, a 3D up-convolutional decoder and a perspective transformer networks. The 2D convolutional encoder consists of 3 convolution layers, followed by 3 fully-connected layers (convolution layers have 64, 128 and 256 channels with fixed filter size of $5 \times 5$; the three fully-connected layers have 1024, 1024 and 512 neurons, respectively). The 3D convolutional decoder

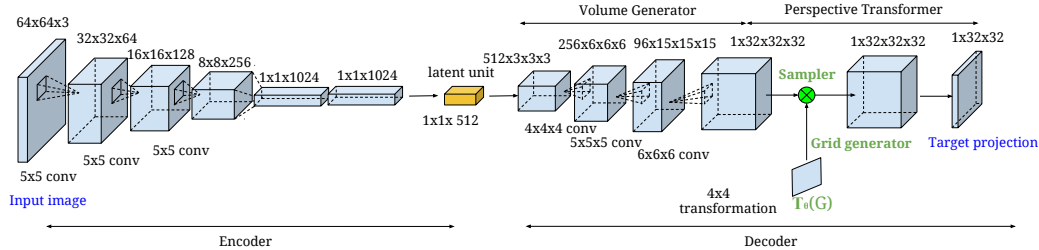

Figure 2: Illustration of network architecture.

consists of one fully-connected layer, followed by 3 convolution layers (the fully-connected layer have $3 \times 3 \times 3 \times 512$ neurons; convolution layers have 256, 96 and 1 channels with filter size of $4 \times 4 \times 4$, $5 \times 5 \times 5$ and $6 \times 6 \times 6$). For perspective transformer networks, we used perspective transformation to project 3D volume to 2D silhouette where the transformation matrix is parametrized by 16 variables and sampling grid is set to $32 \times 32 \times 32$. We use the same network architecture for all the experiments.

**Implementation Details.** We used the ADAM [7] solver for stochastic optimization in all the experiments. During the pre-training stage (for encoder), we used mini-batch of size 32, 32, 8, 4, 3 and 2 for training the RNN-1, RNN-2, RNN-4, RNN-8, RNN-12 and RNN-16 as used in Yang et al. [23]. We used the learning rate $10^{-4}$ for RNN-1, and $10^{-5}$ for the rest of recurrent neural networks. During the fine-tuning stage (for volume decoder), we used mini-batch of size 6 and learning rate $10^{-4}$. For each object in a mini-batch, we include projections from all 24 views as supervision. The models including the perspective transformer nets are implemented using Torch [3]. To download the code, please refer to the project webpage: `http://goo.gl/YEJ2H6`.

**Experimental Design.** As mentioned in the formulation, there are several variants of the model depending on the hyper-parameters of learning objectives $\lambda_{proj}$ and $\lambda_{vol}$. In the experimental section, we denote the model trained with projection loss only, volume loss only, and combined loss as **PTN-Proj** (PR), **CNN-Vol** (VO), and **PTN-Comb** (CO), respectively.

In the experiments, we address the following questions: (1) Will the model trained with combined loss achieve better single-view 3D reconstruction performance over model trained on volume loss only (PTN-Comb vs. CNN-Vol)? (2) What is the performance gap between the models with and without ground-truth volumes (PTN-Comb vs. PTN-Proj)? (3) How do the three models generalize to instances from unseen categories which are not present in the training set? To answer the questions, we trained the three models under two experimental settings: single category and multiple categories.

## 4.1 Training on single category

We select `chair` category as the training set for single category experiment. For model comparisons, we first conduct quantitative evaluations on the generated 3D volumes from the test set single-view images. For each instance in the test set, we generate one volume per view image (24 volumes generated in total). Given a pair of ground-truth volume and our generated volume (threshold is 0.5), we computed its intersection-over-union (IU) score and the average IU score is calculated over 24 volumes of all the instances in the test set. In addition, we provide a baseline method based on nearest neighbor (NN) search. Specifically, for each of the test image, we extract VGG feature from `fc6` layer (4096-dim vector) [17] and retrieve the nearest training example using Euclidean distance in the feature space. The ground-truth 3D volume corresponds to the nearest training example is naturally regarded as the retrieval result.

Table 1: Prediction IU using the models trained on `chair` category. Below, "`chair`" corresponds to the setting where each object is observable with full azimuth angles, while "`chair-N`" corresponds to the setting where each object is only observable with narrow range (subset) of azimuth angles.

| Method / Evaluation Set | chair | | chair-N | |
|---|---|---|---|---|
| | training | test | training | test |
| PTN-Proj:single (no vol. supervision) | 0.5712 | 0.5027 | 0.4882 | **0.4583** |
| PTN-Comb:single (vol. supervision) | **0.6435** | **0.5067** | **0.5564** | 0.4429 |
| CNN-Vol:single (vol. supervision) | 0.6390 | 0.4983 | 0.5518 | 0.4380 |
| NN search (vol. supervision) | — | 0.3557 | — | 0.3073 |

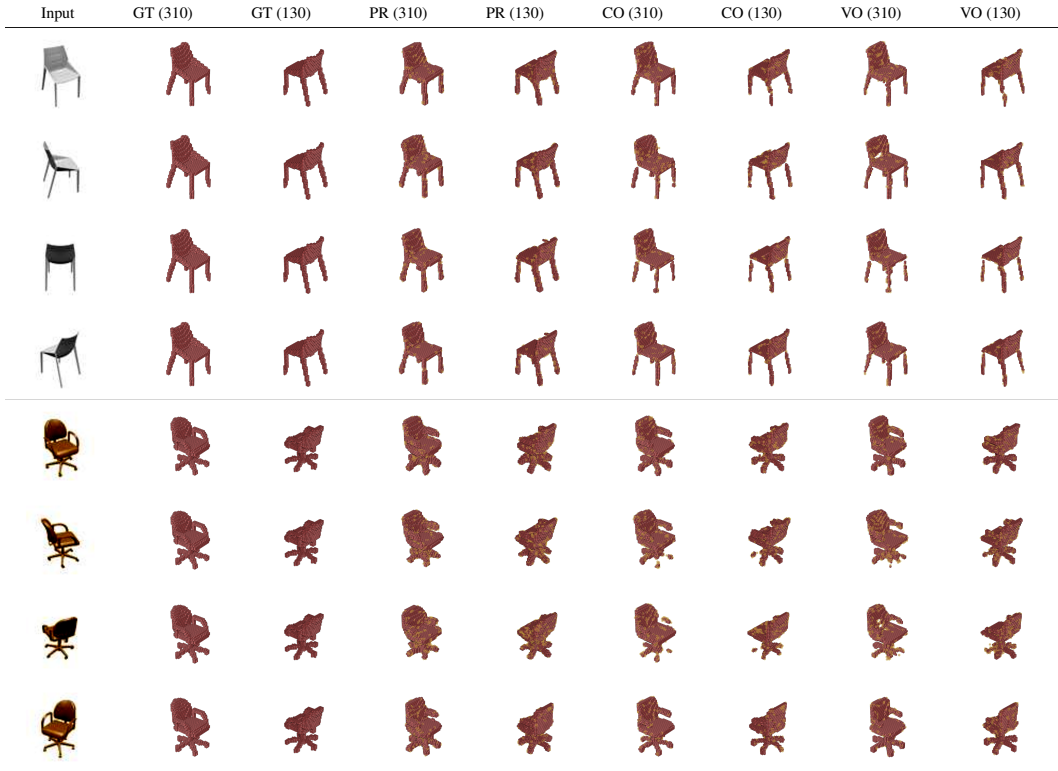

| Input | GT (310) | GT (130) | PR (310) | PR (130) | CO (310) | CO (130) | VO (310) | VO (130) |
|-------|----------|----------|----------|----------|----------|----------|----------|----------|

Figure 3: Single-class results. GT: ground truth, PR: PTN-Proj, CO: PTN-Comb, VO: CNN-Vol (Best viewed in digital version. Zoom in for the 3D shape details). The angles are shown in the parenthesis. Please also see more examples and video animations on the project webpage.

As shown in Table 1, the model trained without volume supervision (projection loss) performs as good as model trained with volume supervision (volume loss) on the `chair` category (testing set). In addition to the comparisons of overall IU, we measured the view-dependent IU for each model. As shown in Figure 4, the average prediction error (mean IU) changes as we gradually move from the first view to the last view ($15°$ to $360°$). For visual comparisons, we provide a side-by-side analysis for each of the three models we trained. As shown in Figure 3, each row shows an independent comparison. The first column is the 2D image we used as input of the model. The second and third column show the ground-truth 3D volume (same volume rendered from two views for better visualization purpose). Similarly, we list the model trained with projection loss only (PTN-Proj), combined loss (PTN-Comb) and volume loss only (CNN-Vol) from fourth column up to ninth column. The volumes predicted by PTN-Proj and PTN-Comb faithfully represent the shape. However, the volumes predicted by CNN-Vol do not form a solid chair shape in some cases.

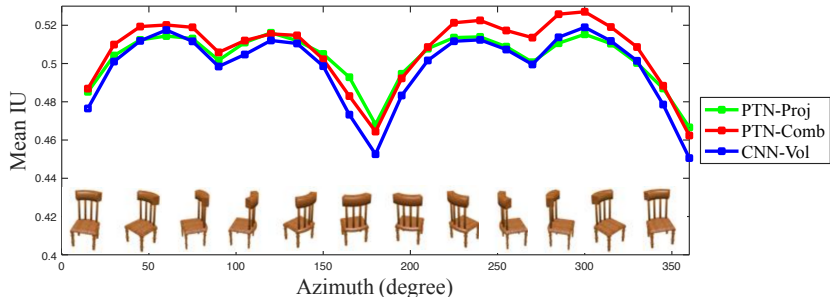

Figure 4: View-dependent IU. For illustration, images of a sample chair with corresponding azimuth angles are shown below the curves. For example, 3D reconstruction from $0°$ is more difficult than from $30°$ due to self-occlusion.

Table 2: Prediction IU using the models trained on large-scale datasets.

| Test Category | airplane | bench | dresser | car | chair | display | lamp |
|---|---|---|---|---|---|---|---|
| PTN-Proj:multi | 0.5556 | 0.4924 | 0.6823 | 0.7123 | 0.4494 | 0.5395 | **0.4223** |
| PTN-Comb:multi | **0.5836** | 0.5079 | **0.7109** | **0.7381** | **0.4702** | **0.5473** | 0.4158 |
| CNN-Vol:multi | 0.5747 | **0.5142** | 0.6975 | 0.7348 | 0.4451 | 0.5390 | 0.3865 |
| NN search | 0.5564 | 0.4875 | 0.5713 | 0.6519 | 0.3512 | 0.3958 | 0.2905 |

| Test Category | loudspeaker | rifle | sofa | table | telephone | vessel |
|---|---|---|---|---|---|---|
| PTN-Proj:multi | **0.5868** | 0.5987 | 0.6221 | 0.4938 | 0.7504 | **0.5507** |
| PTN-Comb:multi | 0.5675 | **0.6097** | **0.6534** | **0.5146** | **0.7728** | 0.5399 |
| CNN-Vol:multi | 0.5478 | 0.6031 | 0.6467 | 0.5136 | 0.7692 | 0.5445 |
| NN search | 0.4600 | 0.5133 | 0.5314 | 0.3097 | 0.6696 | 0.4078 |

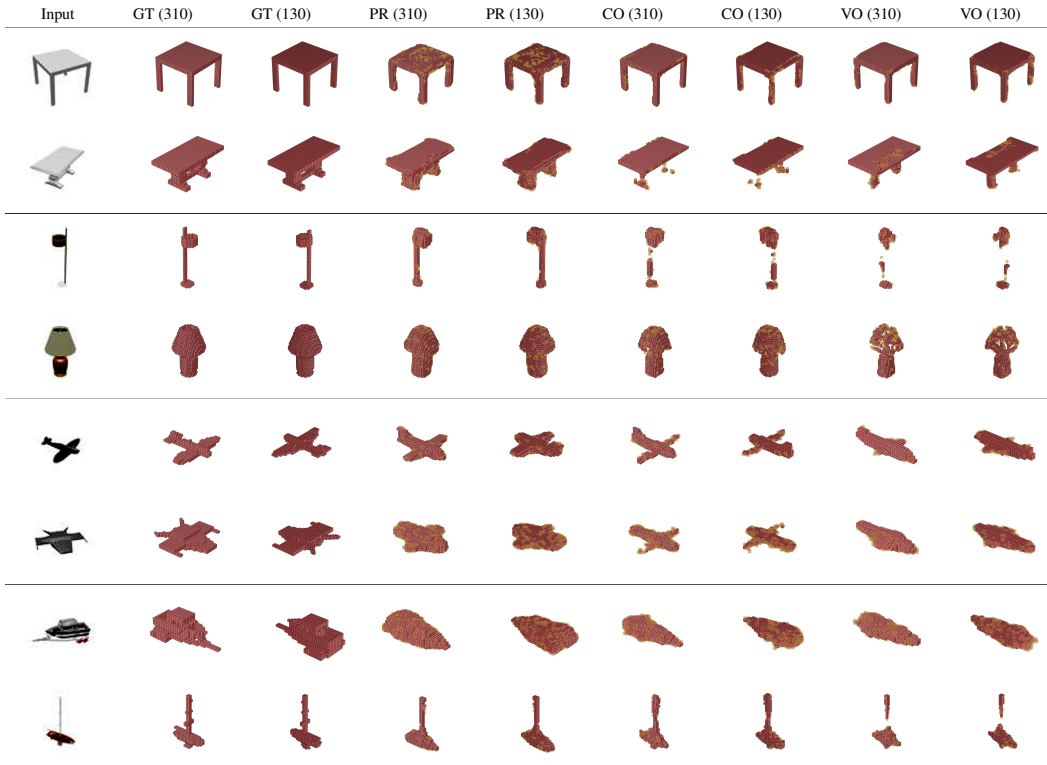

Figure 5: Multiclass results. GT: ground truth, PR: PTN-Proj, CO: PTN-Comb, VO: CNN-Vol (Best viewed in digital version. Zoom in for the 3D shape details). The angles are shown in the parenthesis. Please also see more examples and video animations on the project webpage.

**Training with partial views.** We also conduct control experiments where each object is only observable from narrow range of azimuth angles (e.g., 8 out of 24 views such as $0°$, $15°$, $\cdots$, $105°$). We include the detailed description in the supplementary materials. As shown in Table 1 (last two columns), performances of all three models drop a little bit but the conclusion is similar: the proposed network (1) learns better 3D shape with projection regularization and (2) is capable of learning the 3D shape by providing 2D observations only.

## 4.2 Training on multiple categories

We conducted multiclass experiment using the same setup in the single-class experiment. For multi-category experiment, the training set includes 13 major categories: airplane, bench, dresser, car, chair, display, lamp, loudspeaker, rifle, sofa, table, telephone and vessel. Basically, we preserved 20% of instances from each category as testing data. As shown in Table 2, the quantitative results demonstrate (1) model trained with combined loss is superior to volume loss in most cases and (2) model trained with projection loss perform as good as volume/combined loss. From the visualization results shown in Figure 5, all three models predict volumes reasonably well. There is only subtle performance difference in object part such as the wing of airplane.

Table 3: Prediction IU in out-of-category tests.

| Method / Test Category | bed | bookshelf | cabinet | motorbike | train |
|---|---|---|---|---|---|
| PTN-Proj:single (no vol. supervision) | **0.1801** | **0.1707** | **0.3937** | **0.1189** | **0.1550** |
| PTN-Comb:single (vol. supervision) | 0.1507 | 0.1186 | 0.2626 | 0.0643 | 0.1044 |
| CNN-Vol:single (vol. supervision) | 0.1558 | 0.1183 | 0.2588 | 0.0580 | 0.0956 |
| PTN-Proj:multi (no vol. supervision) | **0.1944** | **0.3448** | **0.6484** | **0.3216** | 0.3670 |
| PTN-Comb:multi (vol. supervision) | 0.1647 | 0.3195 | 0.5257 | 0.1914 | **0.3744** |
| CNN-Vol:multi (vol. supervision) | 0.1586 | 0.3037 | 0.4977 | 0.2253 | 0.3740 |

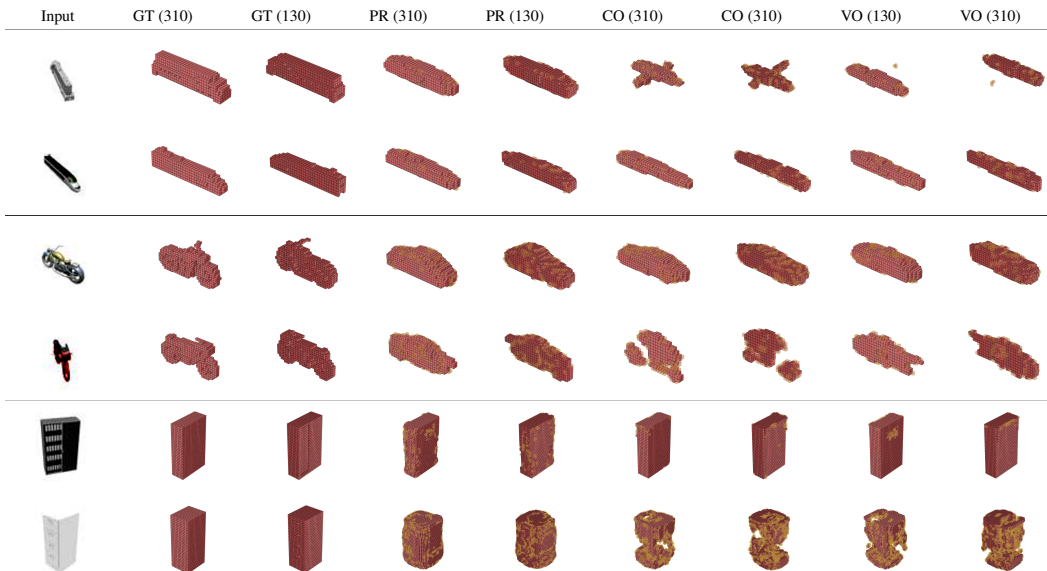

Figure 6: Out-of-category results. GT: ground truth, PR: PTN-Proj, CO: PTN-Comb, VO: CNN-Vol (Best viewed in digital version. Zoom in for the 3D shape details). The angles are shown in the parenthesis. Please also see more examples and video animations on the project webpage.

## 4.3 Out-of-Category Tests

Ideally, an intelligent agent should have the ability to generalize the knowledge learned from previously seen categories to unseen categories. To this end, we design out-of-category tests for both models trained on a single category and multiple categories, as described in Section 4.1 and Section 4.2, respectively. We select 5 unseen categories from ShapeNetCore: bed, bookshelf, cabinet, motorbike and train for out-of-category tests. Here, the two categories cabinet and train are relatively easier than other categories since there might be instances in the training set with similar shapes (e.g., dresser, vessel, and airplane). But the bed,bookshelf and motorbike can be considered as completely novel categories in terms of shape.

We summarized the quantitative results in Table 3. Suprisingly, the model trained on multiple categories still achieves reasonably good overall IU. As shown in Figure 6, the proposed projection loss generalizes better than model trained using combined loss or volume loss on train, motorbike and cabinet. The observations from the out-of-category tests suggest that (1) generalization from a single category is very challenging, but training from multiple categories can significantly improve generalization, and (2) the projection regularization can help learning a robust representation for better generalization on unseen categories.

## 5 Conclusions

In this paper, we investigate the problem of single-view 3D shape reconstruction from a learning agent's perspective. By formulating the learning procedure as the interaction between 3D shape and 2D observation, we propose to learn an encoder-decoder network which takes advantage of the projection transformation as regularization. Experimental results demonstrate (1) excellent performance of the proposed model in reconstructing the object even without ground-truth 3D volume as supervision and (2) the generalization potential of the proposed model to unseen categories.

**Acknowledgments**

This work was supported in part by NSF CAREER IIS-1453651, ONR N00014-13-1-0762, Sloan Research Fellowship, and a gift from Adobe. We acknowledge NVIDIA for the donation of GPUs. We also thank Yuting Zhang, Scott Reed, Junhyuk Oh, Ruben Villegas, Seunghoon Hong, Wenling Shang, Kibok Lee, Lajanugen Logeswaran, Rui Zhang and Yi Zhang for helpful comments and discussions.

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
