[Supplementary Material · nips2016_935_supp.pdf]

# Perspective Transformer Nets: Learning Single-View 3D Object Reconstruction without 3D Supervision Supplementary Materials

**Xinchen Yan**[1]  **Jimei Yang**[2]  **Ersin Yumer**[2]  **Yijie Guo**[1]  **Honglak Lee**[1,3]
[1]University of Michigan, Ann Arbor
[2]Adobe Research
[3]Google Brain
{xcyan,guoyijie,honglak}@umich.edu, {jimyang,yumer}@adobe.com

## 1 Details regarding perspective transformer network

As defined in the main text, 2D silhouette $S^{(k)}$ is obtained via perspective transformation given input 3D volume $\mathbf{V}$ and specific camera viewpoint $\alpha^{(k)}$.

**Perspective Projection.** In this work, we implement the perspective projection (see Figure 1) with a 4-by-4 transformation matrix $\mathbf{\Theta}_{4\times 4}$, where $\mathbf{K}$ is camera calibration matrix and $(\mathbf{R}, \mathbf{t})$ is extrinsic parameters.

$$\mathbf{\Theta}_{4\times 4} = \begin{bmatrix} \mathbf{K} & 0 \\ 0^T & 1 \end{bmatrix} \begin{bmatrix} \mathbf{R} & \mathbf{t} \\ 0^T & 1 \end{bmatrix} \tag{1}$$

For each point $\mathbf{p}_i^s = (x_i^s, y_i^s, z_i^s, 1)$ in 3D world coordinates, we compute the corresponding point $\mathbf{p}_i^t = (x_i^t, y_i^t, 1, d_i^t)$ in screen coordinates (plus disparity $d_i^t$) using the perspective transformation: $\mathbf{p}_i^s \sim \mathbf{\Theta}_{4\times 4} \mathbf{p}_i^t$.

Figure 1: Illustration of perspective projection. The minimum and maximum disparity in the screen coordinates are denoted as $d_{min}$ and $d_{max}$

Similar to the spatial transformer network introduced in [1], we propose a 2-step procedure: (1) performing dense sampling from input volume (in 3D world coordinates) to output volume (in screen coordinates), and (2) flattening the 3D spatial output across disparity dimension. In the experiment, we assume that transformation matrix is always given as input, parametrized by the viewpoint $\alpha$. Again, the 3D point $(x_i^s, y_i^s, z_i^s)$ in input volume $\mathbf{V} \in \mathbb{R}^{H \times W \times D}$ and corresponding point $(x_i^t, y_i^t, d_i^t)$ in output volume $\mathbf{U} \in \mathbb{R}^{H' \times W' \times D'}$ is linked by perspective transformation matrix $\mathbf{\Theta}_{4\times 4}$. Here, $(W, H, D)$ and $(W', H', D')$ are the width, height and depth of input and output volume, respectively.

$$\begin{pmatrix} x_i^s \\ y_i^s \\ z_i^s \\ 1 \end{pmatrix} = \begin{bmatrix} \theta_{11} & \theta_{12} & \theta_{13} & \theta_{14} \\ \theta_{21} & \theta_{22} & \theta_{23} & \theta_{24} \\ \theta_{31} & \theta_{32} & \theta_{33} & \theta_{34} \\ \theta_{41} & \theta_{42} & \theta_{43} & \theta_{44} \end{bmatrix} \begin{pmatrix} \tilde{x_i}^t \\ \tilde{y_i}^t \\ \tilde{z_i}^t \\ 1 \end{pmatrix} \tag{2}$$

In addition, we compute the normalized coordinates by $x_i^t = \frac{\tilde{x}_i^t}{\tilde{z}_i^t}$, $y_i^t = \frac{\tilde{y}_i^t}{\tilde{z}_i^t}$ and $d_i^t = \frac{1}{\tilde{z}_i^t}$, where $d_i$ is the disparity.

**Differentiable Volume Sampling.** To perform transformation from input volume to output volume, we adopt the similar sampling strategy as proposed in [1]. That is, each point $(x_i^s, y_i^s, z_i^s)$ defines a spatial location where a sampling kernel $k(\cdot)$ is applied to get the value at a particular voxel in the output volume $U$.

$$U_i = \sum_n^H \sum_m^W \sum_l^D V_{nml} k(x_i^s - m; \Phi_x) k(y_i^s - n; \Phi_y) k(z_i^s - l; \Phi_z) \quad \forall i \in \{1, ..., H'W'D'\} \quad (3)$$

Here, $\Phi_x$, $\Phi_y$ and $\Phi_z$ are parameters of a generic sampling kernel $k(\cdot)$ which defines the interpolation method. We implement bilinear sampling kernel $k(x) = \max(0, 1 - |x|)$ in this work.

Finally, we summarize the dense sampling step and channel-wise flattening step as follows.

$$U_i = \sum_n^H \sum_m^W \sum_l^D V_{nml} \max(0, 1 - |x_i^s - m|) \max(0, 1 - |y_i^s - n|) \max(0, 1 - |z_i^s - l|)$$
$$(4)$$
$$S_{n'm'} = \max_{l'} U_{n'm'l'}$$

Note that we use the $\max$ operator for projection instead of summation along one dimension since the volume is represented as a binary cube where the solid voxels have value 1 and empty voxels have value 0. Intuitively, we have the following two observations: (1) each empty voxel will not contribute to the foreground pixel of $S$ from any viewpoint; (2) each solid voxel can contribute to the foreground pixel of $S$ only if it is visible from specific viewpoint.

## 2   Details regarding learning from partial views

In our experiments, we have access to 2D projections from the entire 24 azimuth angles for each object in the training set. A natural but more challenging setting is to learn 3D reconstruction given only partial views for each object. To evaluate the performance gap of using partial views during training, we train the model in two different ways: 1) using narrow range of azimuths and 2) using sparse azimuths. For the first one, we constrain the azimuth range of $105°$ (8 out of 24 views). For the second one, we provide 8 views which form full $360°$ rotation but with a larger stepsize of $45°$.

For both tasks, we conduct the training based on the new constraints. More specifically, we pre-train the encoder using the method proposed by Yang et al. [2] with similar curriculum learning strategy: RNN-1, RNN-2, RNN-4 and finally RNN-7 (since only 8 views are available during training). For fine-tuning step, we limit the number of input views based on the constraint. For evaluation in the test set, we use all the views so that the numbers are comparable with original setting. As shown in Table 1, performances of all three models drop a little bit. Overall, the proposed network (1) learns better 3D shape with projection regularization and (2) is capable of learning the 3D shape by providing 2D observations only. Note that the partial view experiments are conducted on single category only, but we believe the results will be consistent in multiple categories.

Table 1: Prediction IU using the models trained on `chair` category. Here, `chair` corresponds to the setting where each object is observable with full azimuth angles; `chair-N` corresponds to the setting where each object is only observable with narrow range of azimuth angles. `chair-S` corresponds to the setting where each object is only observable with sparse azimuth angles.

| Method / Evaluation Set | chair | | chair-N | | chair-S | |
|---|---|---|---|---|---|---|
| | training | test | training | test | training | testing |
| PTN-Proj:single | 0.5712 | 0.5027 | 0.4882 | **0.4583** | 0.5201 | **0.4869** |
| PTN-Comb:single | **0.6435** | **0.5067** | **0.5564** | 0.4429 | **0.6037** | 0.4682 |
| CNN-Vol:single | 0.6390 | 0.4983 | 0.5518 | 0.4380 | 0.5712 | 0.4646 |

## 3   Additional visualization results on 3D volumetric shape reconstruction

Figure 2: Singleclass results. GT: ground truth, PR: PTN-Proj, CO: PTN-Comb, VO: CNN-Vol (Best viewed in digital version. Zoom in for the 3D shape details). The angles are shown in the parenthesis. Please also see more examples and video animations on the project webpage.

Figure 3: Multiclass results. GT: ground truth, PR: PTN-Proj, CO: PTN-Comb, VO: CNN-Vol (Best viewed in digital version. Zoom in for the 3D shape details). The angles are shown in the parenthesis. Please also see more examples and video animations on the project webpage.