[Reviews · NeurIPS 2016]

Reviewer 1

Summary

This paper attempts to reconstruct a 3D volume for an object from a single image at test time. During training time it uses a number of views of the object to reconstruct a 3D volume containing the object where the volume is broken down into smaller voxels and the network predicts whether each voxel is occupied or not. The input is an image of the object only against a white background. They chose to ignore color and texture in their reconstruction work. The network they suggest is an encoder-decoder network where one half encodes an images into a 3D invariant latent representation and the decoder does dense reconstruction of only that object. Other networks like VoxNet and MultiView CNN have used the encoding network for object recognition. The new contribution of the paper is the 3D reconstruction part from the encoded representation. They use two losses: projection loss and volumetric loss(if model is available). Projection loss takes the 3d model and the camera parameters and produces a silhouette image which is the input image itself(because they have a clean background with only the object in the center of the input image). Volumetric loss compares the 3d reconstruction(voxel wise) with the ground truth 3d model. Comments: 1) The authors leave the training details to be read in another paper which is kind of annoying as they claim the encoding they will learn will be independent of the view and that part comes from another paper. Also, they need to mention if they use the same network for all the categories or does one need to train a different network for each category. From the paper it seems they have separate networks for separate categories. 2) They use Intersection over Union(IU) of 3D volume as the chosen metric. The numbers from some interpretable simple baseline should be presented for this metric so as to make more sense out of this IU number. They only make comparison with variants of their own model. If an object is big then one might get good number by always predicting the voxel is occupied. The network is supposed to output a fixed volume for all viewpoints of the object. I am curious as to if you took an average model for a category what would the IU numbers be. 3) The volumetric loss degrades performance a bit. This seems to be a little strange as this is the loss function that is directly related to the reconstruction task. The results are pretty close to projection loss though. 4) They fix the elevation angle of the images being rendered to 30 degree at training time. It seems that is the same angle they have in all the test images.

Qualitative Assessment

The main criticism of this work, is the empirical evaluation. The authors do not compare their results to those of other established algorithms, but only to the different versions of the proposed method. Second, There is no explanation of the observed results. For example why is the projected model for busses is worse than the combined, while is the opposed for trains, eventhough the 3d models are extremely similar. Furthermore, the evaluation is biased towards larger objects, yet it is not acknowledged in the text.

Confidence in this Review

3-Expert (read the paper in detail, know the area, quite certain of my opinion)


Reviewer 2

Summary

The submission considers an interesting problem of inferring a 3D shape (volumetric model) of an object from a single silhouette and uses large-scale learning (of deep arhitectures, of course) to achieve this goal. Not surprisingly, some reasonable predictions are obtained when the training set is large and when the shape set is restricted to a certain class (cross-category experiments are also presented but they show the model's inability to generalize). The main algorithmic contribution is the new loss that penalizes deviation between 2D projections (for displaced camera views), as the authors show that such loss leads to better performance compared to straight-forward loss that looks at the mismatch between the prediction and the ground truth in 3D.

Qualitative Assessment

I believe that the paper makes sense and introduces a reasonable idea. There are obvious relations between the proposed loss and the work of Tatarchenko et al.[19] (perhaps worth more lengthy discussion). Furthermore, approaches that do 2.5d reconstruction of objects of a certain class (e.g. [Vicente et al. CVPR14],[Kak et al. CVPR15] and perhaps more face-related approaches) should be referenced and discussed. I believe that the authors incorrectly relate their approach to photo hulls. What they do is actually related to visual hulls (since their work are about silhouettes and does not deal with colors/intensities). When presenting the results (for within-category case), it would be very useful to show the closest training example in each case. This would greatly help to assess the amount of generalization the system is capable of. Finally, figure 6 should indicate the training classes in each case.

Confidence in this Review

2-Confident (read it all; understood it all reasonably well)


Reviewer 3

Summary

The paper provides an approach to learn a latent representation for 3D objects and object classes by means of an encoder-decoder network, with an attached projective transformer network (cf Jaderberg et al.). The main novelty, besides the particular combination of these two networks, is that the loss function does not necessarily rely on 3D ground truth volumes; instead, it can be purely based on a projection loss. In the experiments, different loss functions (volumetric, projection, and combined) are explored, and the superiority of the projection loss is demonstrated. The experiments furthermore show results on single-category and multi-category data, as well as some out-of-category experiments.

Qualitative Assessment

I enjoyed reading the clearly and concisely written submission. The contributions are incrementally building upon previous work, but the novelty of the chosen approach (encoder-decoder network plus spatial transformer network, in combination with the projection loss function) is conveyed convincingly. The experiments clearly show the superiority of the chosen projection (or combined) loss approach, compared to using only the volumetric loss. My main comment is related to the projection loss function: If it is based on space carving theory, the easiest representation for the network to learn during training might be the convex hull of the object (whose volumetric representation is potentially unknown), instead of the actual representation of the object. This of course could explain the view-dependent mean IU scores of Figure 4. On the other hand, it is mentioned in ll. 144-146 that projection from different viewpoints are bundled together in a mini-batch in order to aid learning of consistent representations. It is not entirely clear to me how well unseen concavities in the projection can be or are modeled by the network. It would be great if the authors could spend a few words on this, as well as on explaining the impact of the mini-batch bundling (as opposed to, say, random sampling per mini batch). I am aware that the main focus of the paper is a proof of concept of the main contributions. However, I also wonder how the exact network architecture was chosen for the decoder and encoder parts. Did the authors try changing the architecture to increase its capacity, either giving it more "depth" (e.g. more layers) or more spatial resolution? Was orthogonal projection implemented due to its easier representation compared to perspective projection? Is there any insight whether choosing perspective projection would make any difference? The following reference should should probably be added for completeness: - Dosovitskiy et al., "Learning to Generate Chairs with Convolutional Neural Networks", CVPR 2015. (http://lmb.informatik.uni-freiburg.de/Publications/2015/DB15/Generate_Chairs_arxiv.pdf) Overall, I find the submission of high quality and recommend it for a presentation at NIPS.

Confidence in this Review

2-Confident (read it all; understood it all reasonably well)


Reviewer 4

Summary

The paper proposes an alternative/additional loss for the task of single image object reconstruction - that the projections of the predicted and ground-truth 3D shape should agree across various views. This is cleanly implemented in a CNN framework using spatial transformers and the paper empirically demonstrates the benefits of using this loss instead of / in addition to a conventional loss.

Qualitative Assessment

I like the motivation of the paper that in case full 3D information is not available, we can still train a volume prediction CNN using an alternative loss which enforces silhouette consistency with available observations (eg. can be used on Ebay/Amazon etc.). The implementation proposed in the paper (using spatial transformers and max-projection on predicted volume to obtain a silhouette) is also clean, intuitive and can be easily adoptable. My main concerns are regarding the experimental setup and results (explained in detail below) Experimental Results : A central point in the experiments is that using the projection loss gives better reconstructions as well as more generalizable models - I don't understand why that should be true. Basically, the projection loss is just an alternative loss which instead of directly penalizing distance between predicted and true models, instead penalizes difference in their projections. It is not clear why this a better loss or why it adds any value at all in model-comb (except perhaps as an additional regularizer). However, the experiments show a clear gain over model-vol (chair in Table 1, generalization in Table 2) - one possible explanation for this is the evaluation metric (IU @ 0.5). Since the models are trained independently, the threshold 0.5 is really arbitrary (as different models may be calibrated differently) and I suspect having max-projection and loss with silhouette makes it a better threshold - for eg., the predictions by VO in Fig 3 are not well formed, but they would be if a lower threshold were used. I strongly recommend that the following experiment be run for the rebuttal - instead of measuring IU @ 0.5, the evaluation metric should be IU @ t where the threshold t is selected independently for each model (and category). If the conclusions presented still hold, I would be happy to update my assessment but otherwise I think this is a serious concern as I find it extremely counter-intuitive that the projective loss would help in training better/more general reconstruction models. Experimental Setup : Despite the concerns above, I agree that the projection-loss based strategy is useful as it allows one to train without having the underlying 3D model available. Unfortunately, the experimental setup in the paper does not present this strongly - as L172 states, all the 24 views of the object are used for training the CNN and while technically the full 3D model is not available, it practically is. Infact recent experiments in the community (eg. Multi-view CNN) show that representing a 3D model by a collection of views instead of voxels performs even better for category classification. The point here being that using a diverse set of views essentially has similar information as using the full 3D model. If the experiments were to use a more natural setting where for each 3D model, only a few views from a narrow range of azimuth were available (which is arguably a better model for a learning agent which moves around objects) and still obtain comparable reconstruction, it would be a much stronger demonstration of the utility of the technique presented here. Overall, while the central idea is interesting and novel with a clean implementation, there are concerns regarding both the conclusions mentioned - 1) the experimental setup used here is not really representative of a secnario with 'ground-truth 3D volume not available' and 2) It is unclear if the generalization is better die to the the alternative loss or is an artifact of the metric - hopefully the experiment suggested above would clarify (though the question of why it is better would still remain). Minor Concerns/Suggestions : - Technically, the encoder+decoder is supposed to learn a 3D view invariant output and that should just emerge as training progresses, and it should not be necessary to enforce anything on the encoder. The pre-training described in L134-144 seems adhoc. Also, it's unclear why using the deep-rotator encoder does leads to view-invariance since it's not explicitly enforced during the deep-rotator training. - Using L1 loss instead of L2 in Eqn 2 typically leads to better reconstructions. - While the overall writing of the paper was great, the introduction using the conscious learning agent seems contrived. As mentioned above, an active agent would not explore all diverse views of the car, would not have a common canonical frame across all cars (and so not know the absolute pose of the car - only relative pose of different views) etc. ----- Update after Rebuttal : The authors addressed the major concerns above by showing the conclusions hold with the suggested evaluation. An additional experiment with a restricted set of training views available for each object was also presented and I feel this better justifies the motivation of the paper.

Confidence in this Review

3-Expert (read the paper in detail, know the area, quite certain of my opinion)


Reviewer 5

Summary

This paper presents to a 3D volumetric shape reconstruction method using only a single-view image. This is a very interesting and challenging task in computer vision. The authors formulate this task by finding the intrinsic relationships between 2D and 3D representations.The relationships are learned using an encoder-decoder network with a projection transform loss. The experimental results show that the proposed reconstruction method with projection loss can achive better performance and better generalization ability than other two loss functions: loss with the groud-truth 3D shapes, and the loss of combing both projection and groud-truth 3D shapes.

Qualitative Assessment

-Single-view 3D object reconstruction is a quite challenging task in computer vision. Formulating this difficult task as a network learning problem is quite novel to me. As mentioned by the authors, there is only one very related work (ref. 3) which solve the similar problem (using single image or multiple images) by using a framework of encoder, 3D convolutional LSTM, and decoder. Different from [3], the authors formulate this problem by using the idea of 3D spatial transformer networks proposed in [6].The experimental results demonstrate the effetiveness and the generalization ability of the proposed method.   The main advatange of this method is: it do not need any groud-truth volumentric shape during the training. Thus, this work has some notable novelty and contributions. -In the experiments, the authors demostrate their results by training the network on single category and multiple categories. The results based on different loss functions are compared. However, the results of the most related work mentioned in reference [3] are not compared. By my visual comparison, I found that some results in [3] look better than the ones in your paper. For example, when the object (e.g.,bicycle) has some topological holes, the proposed method may fail. Could you please explain the reasons? On the other hand, could you please give more explainations on why in most cases the results achived by using only the projection loss are better than the ones adding the groud-truth volumetric information? - Since the 3D shape reconstruction from 2D image(s) is a fundamental problem in computer vision, this work or the learning style used in this work will have considerabale impact in the communite. - The whole paper, inculding the model, algorithm and experiments, is well presented. The experimental results, inculding the video demo in the supplement file are interesting to me. Overall, this paper has some notable novel contributions with good presentations. I suggest to accept it.

Confidence in this Review

2-Confident (read it all; understood it all reasonably well)